# How Do Specific Environmental Conditions in Canals Affect the Structure and Variability of the Zooplankton Community?

Nikola Kolarova and Paweł Napiórkowski *

Department of Hydrobiology, Faculty of Biological Sciences, Kazimierz Wielki University, Ossollińskich 12 Street, 85-093 Bydgoszcz, Poland; nikol77@student.ukw.edu.pl
* Correspondence: pnapiork@ukw.edu.pl

**Abstract:** The present study investigates the responses of zooplankton (including changes in their structure and diversity) to physicochemical and biological parameters in two artificial waterways. Water samples were collected monthly from the Bydgoszcz Canal, the Noteć Canal, and the Brda River during the growing season of April–October 2019. We analyzed how selected parameters (including water temperature, Secchi disk visibility, oxygen concentration, conductivity, and pH, as well as nitrate, phosphate, and chl-a concentrations) affected seasonal variations in zooplankton diversity (T) and density (N). In total, we recorded 98 species, and average zooplankton density was 320 ind/L. At all sites, the same zooplankton species were dominant: *Keratella cochlearis* among rotifers and the Cladocera *Bosmina longirostris* among crustaceans. Rotifers dominated qualitatively and quantitatively over crustaceans. Zooplankton density and biomass, as well as the number of zooplankton species, were higher in the Bydgoszcz Canal than in the Brda River or the Noteć Canal. This may be connected to the locks on the Bydgoszcz Canal slowing down water flow, thereby increasing macrophyte vegetation, which creates ecological niches supporting zooplankton development.

**Keywords:** rotifers; crustaceans; Bydgoszcz Canal; Brda River; hydrological conditions; water parameters

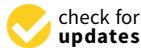



## 1. Introduction

Canals have an important hydrological function: they connect rivers to form a large inland water system [1]. Similar to natural waterways, they are characterized by varied hydrological regimes and environmental conditions [2–4]. The main criteria for determining these conditions are species diversity and the quantitative ratio and biological productivity of species [5]. Canals can play an important role in transporting alien and invasive species, including planktonic organisms [6]. Because of diverse hydrological conditions, these structures may provide different environmental conditions suitable for aquatic biota; for example, low water flow reduces zooplankton movement, thereby supporting their reproduction and growth [7–9]. The zooplankton community is an important element of aquatic ecosystems [10]. These organisms respond rapidly to changes in the environment and thus help determine changes in water quality [11]. Therefore, productivity in aquatic ecosystems is correlated with the zooplankton community structure [12]. Zooplankton diversity and density are essential to keep the ecosystem healthy because each species may have a different effect on ecosystem functioning [13]. Zooplankton depletion in rivers and other waterways may be related to hydrological conditions, mainly to river regime [14,15]. Several authors have observed a significant impact of biotic competition and fish predation on zooplankton structure [9,16]. Zooplankton decline may affect the quality of an aquatic ecosystem and lead to shifts in the trophic chain [17,18]. A species or a group of species may appear to be functionally redundant in certain environmental conditions but may have a different effect on an ecosystem in other conditions [19]. Rotifers and crustaceans

can be used to determine trophic levels in freshwater ecosystems. Zooplankton are highly sensitive to changes in water quality. As a result, spatial variations in zooplankton diversity (T) and density (N) can be determined [20,21]. Depending on changes in environmental parameters connected to shifts in hydrological regimes or disturbances to the river system, zooplankton species composition can differ markedly between different sampling sites in the same river or canal [22]. To date, the hydrobiology of canals of Eastern and Western Europe has not been thoroughly investigated. However, in recent decades, the Bydgoszcz Canal has frequently been studied as a pathway for invasive species [23]. Some studies about zooplankton species in relation to environmental variables have been carried out [11,24–27]. It has been demonstrated that seasonal and spatial variations play an important role in the fluctuations of the environmental parameters that shape the species composition and abundance of zooplankton.

The aim of the study was to compare the zooplankton species composition in two artificial waterways against that of a natural river. We hypothesized that spatial community structure during the growing season would depend on differences in hydrological, environmental, and biological conditions and their influence on food availability (algal growth) and on the creation of ecological niches for zooplankton (macrophytes growth). Specifically, we expected that crustacean diversity (density and species number) would be lower because their development could be disturbed by excessive water flow and competition with rotifers for algal food [28–30].

## 2. Materials and Methods

The study was conducted during the growing season from April to October 2019 in the Bydgoszcz Canal (part of which is located in the industrial area of Bydgoszcz city), the Noteć Canal (located in the rural area near the town of Nakło), and in the Brda River (the sampling site was located near its conjunction with the Bydgoszcz Canal) (Figure 1). The Bydgoszcz Canal is located between the city of Bydgoszcz and the town of Nakło in north-west Poland. It is a very important artificial waterway, part of the E70 international waterway connecting the two largest rivers in Poland (the Vistula and the Oder) through their tributaries, the Brda, Noteć, and Warta rivers. Its total length is 24.7 km, of which 15.7 km lies within the catchment of the Noteć (a tributary of the Oder) and 9.0 km within the catchment of the Brda (a tributary of the Vistula). It is supplied by the Upper Noteć and by small watercourses and streams within Bydgoszcz and the Bydgoszcz–Nakło valley. The canal also collects rainwater, and water from a nearby sewage treatment plant. In recent decades, the canal has received pollution from point and non-point sources in this urban area. Because the function of the canal has changed and its dredging has been suspended, it has become shallower [31]. The Bydgoszcz Canal has six navigation locks along its length. The drops at the locks vary from 1.81 m (the Józefinki lock) to 7.58 m (the Okole and Czyżkówko locks). The depth of the Bydgoszcz Canal ranged from 1.6 to 2.0 m depending on the water level, and the width at different sites ranged from 28 to 30 m (water flow was c. 0.04 m/s). At the locks, in summer (June, July, August), the canal was almost completely covered with floating vegetation and partly with submerged vegetation (fine duckweed, *Lemna minor* L.; star duckweed, *Lemna trisulca* L.; rigid hornwort, *Ceratophyllum demersum* L.; pondweed species, *Potamogeton* sp.). The entire Noteć Canal, consisting of two sections (one simply referred to as the Noteć Canal and the other as the Upper Noteć Canal), is a waterway covering the course of the Noteć River. It has a total length of 25 km and low water discharge and is strongly affected by anthropogenic contaminants due to human activities, including agriculture. The depth of the Noteć Canal ranged from 0.8 to 1.2 m, and the width was about 15 m at the studied site (water flow was c. 0.25 m/s). The Noteć Canal has six navigation locks. Samples were taken at the last lock before the mouth of the Noteć Canal into the Bydgoszcz Canal. The main current of the Noteć Canal was divided into two parts. One was constantly flowing through the turbines of small hydroelectric power plants, while the other was slightly slowing in front the locks. The Brda River, a tributary of the Vistula, is located in northwest Poland and has a length of 245 km and a

catchment area of about 4634 km$^2$. The river flows into Bydgoszcz from the north and is a natural waterway with a width of 20–30 meters until its confluence with the Bydgoszcz Canal [32]. The width of the Brda River at the studied site was 45 m and the depth was *c.*2.5 m (water flow was *c.*0.80 m/s).

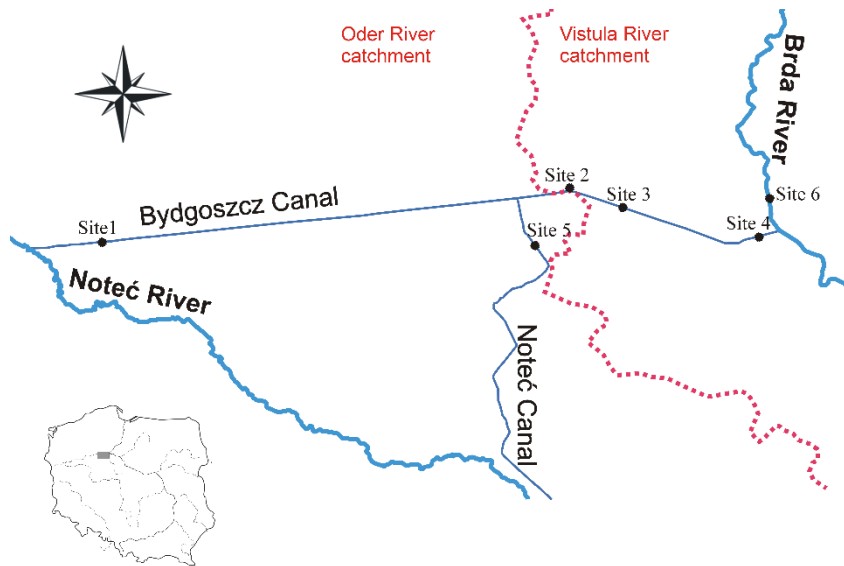

**Figure 1.** Map of investigated area. Bydgoszcz Canal: site 1—Józefinki in front of the sluice; site 2—Osowa Góra in front of the sluice; site 3—Prądy in front of the sluice; site 4—Okole behind the sluice; site 5—Noteć Canal in front of the sluice; site 6—Brda River.

The water flow was measured by electromagnetic hydrometric mill (Model 801) during the study period. The electromagnetic meter measured the water flow twice per 5 s interval. Data were obtained by averaging for this period. Water samples were collected at six sampling sites in three areas: Area 1: the Bydgoszcz Canal: (site 1) Józefinki in front of the sluice 53°07'49.7" N 17°38'23.9" E, (site 2) Osowa Góra in front of the sluice 53°08'48.9" N 17°52'49.2" E, (site 3) Prądy in front of the sluice 53°08'38.6" N 17°53'37.8" E, (site 4) Okole behind the sluice 53°08'11.9" N 17°58'06.1" E; Area 2: (site 5) the Noteć Canal in front of sluice 53°07'56.5" N 17°51'18.1" E; and Area 3: (site 6) the Brda River 53°08'16.0" N 17°58'20.8" E. Samples (for zooplankton and water quality examination) were collected monthly from April to October with a 1 L Patalas bucket at a depth of *c.* 0.5 m. Plankton was collected by filtering. To obtain one sample of zooplankton, 10 L of water was filtered through a plankton net with mesh diameter of *c.* 25 μm. All samples were preserved with Lugol's solution [33,34]. Altogether, 42 water samples were collected. Zooplankton were identified and measured using an Olympus BX 43 light microscope, as well as an Olympus LC 30 soft imaging camera. The samples were prepared for counting under a microscope and the method of counting previously described in [35] was used. Zooplankton density and biomass were calculated per 1 L of water. The commonly available keys were used for taxonomical identification of zooplankton [34,36–39]. The Shannon α-diversity index (H') and Pielou evenness index (J') were used to describe the abundance–dominance relationship. At the same time as samples were collected, the following environmental parameters of water were measured: water temperature (WT, °C), Secchi disk visibility (SD, m), conductivity (EC, μS cm$^{-1}$), oxygen concentration (DO, mg/L), chlorophyll (chl-a, μg/L), nitrates (NNO$_3^-$, mg/L), phosphates (PPO$_4^{2-}$, mg/L), and pH (Table 1). Multimeter WTW Multi 3430SET F Xylem Analytics field probes (Weilheim, Germany) were used for measurements.

**Table 1.** Environmental data during the growing season 2019 in the Bydgoszcz Canal, the Noteć Canal, and the Brda River sites.

| | Bydgoszcz Canal | | | | | Noteć Canal | | Brda River | |
|---|---|---|---|---|---|---|---|---|---|
| | Site 1 | Site 2 | Site 3 | Site 4 | Mean-Range | Site 5 Mean | Range | Site 6 Mean | Range |
| WT (°C) | 16.4 | 16.7 | 17.0 | 16.5 | 16.7 (7.6–23.0) | 16.6 | (7.6–25.8) | 17.0 | (9.2–23.4) |
| SD (m) | 1.05 | 1.29 | 1.26 | 0.84 | 1.11 (0.45–2.10) | 1.11 | (0.45–1.50) | 2.20 | (2.00–2.50) |
| EC ($\mu$S cm$^{-1}$) | 623 | 674 | 586 | 391 | 569 (152–1115) | 672 | (525–1120) | 122 | (101–202) |
| DO (mg/L) | 8.3 | 9.5 | 9.4 | 10.0 | 9.3 (2.7–16.3) | 6.8 | (1.9–12.1) | 8.8 | (6.6–14.2) |
| pH | 7.4 | 7.5 | 7.6 | 7.7 | 7.6 (6.7–8.9) | 7.5 | (6.5–8.3) | 8.0 | (7.0–9.5) |
| chl-a ($\mu$g/L) | 13.46 | 10.44 | 12.42 | 8.39 | 11.18 (0.30 31.31) | 15.78 | (0.92–51.48) | 5.42 | (0–23.81) |
| NNO$_3^-$ (mg/L) | 0.55 | 0.59 | 0.57 | 0.47 | 0.54 (0.15–1.41) | 0.73 | (0.28–1.67) | 0.31 | (0.21–0.49) |
| PPO$_4^{2-}$ (mg/L) | 0.16 | 0.10 | 0.15 | 0.28 | 0.14 (0.02–0.63) | 0.21 | (0.03–0.89) | 0.04 | (0.01–0.07) |

The environmental variables responsible for variations in the zooplankton taxonomic composition, density, and biomass during the growing season at the investigated sites were determined by canonical correspondence analysis (CCA) [40]. The explanatory response variables used in the analyses were WT, SD, pH, DO, EC, chl-a, NNO$_3^-$, and PPO$_4^{2-}$, as well as number of zooplankton species (including total number of zooplankton species, number of rotifer and crustacean species), zooplankton density (including total zooplankton density, rotifer and crustacean density), and zooplankton biomass (including total zooplankton biomass, rotifer and crustacean biomass). The statistical analysis was performed using all data (environmental and biological), including all investigated months. Log (x + 1) transformation was applied before CCA to reduce the influence of outliers on the results. The CCA ordination plots examined the relationships between selected environmental variables and biological data. Only environmental variables explaining significant amounts of variance ($p < 0.01$) and ($p < 0.05$) were retained in the models and tested for significance. The CCA statistical analysis was carried out by Past 4.03 software [41]. Statistically significant correlations between environmental and biological parameters were tested by Spearman's rho using Past 4.03 software [41]. Two-way cluster analysis was performed to group sites based on their similarity within environmental and biological data in the investigated months. All data (environmental and biological) were log (x + 1)-transformed to reduce the influence of outliers on the results. Ward's clustering method and Euclidean distance in PC-ORD 6.08 [42] were used to compare spatial and seasonal similarity of environmental and biological parameters during the study period.

## 3. Results

### 3.1. Taxonomic Diversity and Density

In total, we recorded 98 zooplankton species, including 73 rotifer species (i.e., 75% of all species) and 25 crustacean species (i.e., 25% of all species) alongside nauplii and copepodites, both of which are larval forms of Copepoda. The highest number of species in total was recorded at site 4 in the Bydgoszcz Canal (58), comprising 45 rotifer species and 13 crustacean species (Table 2, Figure 2). The lowest number of species (43) was recorded in the Noteć Cana at site 5, comprising 32 rotifer species and 11 crustacean species. The highest number of species in one sample (both rotifers and crustaceans) was 18, which was also recorded in the Bydgoszcz Canal, as compared to the 15 in the Noteć Canal and 14 in the Brda River. The highest total number of rotifer species was observed in the Bydgoszcz Canal at site 2 (45) and at site 4 (45), and the highest number of crustaceans at site 3 (15) (Table 2).

**Table 2.** Total number of species (diversity) and dominants in the zooplankton community during growing season 2019 in the Bydgoszcz Canal, the Noteć Canal, and the Brda River sites (* Rotifer, ** Crustacean).

| | Bydgoszcz Canal | | | | Noteć Canal | Brda River |
|---|---|---|---|---|---|---|
| | Site 1 | Site 2 | Site 3 | Site 4 | Site 5 | Site 6 |
| Rotifers | 39 | 45 | 40 | 45 | 32 | 38 |
| Crustaceans | 12 | 12 | 15 | 13 | 11 | 11 |
| Total | 51 | 57 | 55 | 58 | 43 | 49 |
| Dominant species and percent of domination | *Keratella cochlearis* * 62% | *Anuraeopsis fissa* * 33% | *Keratella cochlearis* * 61% | *Keratella cochlearis* * 29% | *Keratella cochlearis* * 60% | *Keratella cochlearis* * 26% |
| | *Keratella quadrata* * 6% | *Keratella cochlearis* * 6% | *Keratella quadrata* * 7% | *Keratella quadrata* * 18% | *Keratella quadrata* * 14% | *Keratella quadrata* * 23% |
| | *Polyarthra remata* * 8% | *Polyarthra dolichoptera* * 21% | *Polyarthra dolichoptera* * 20% | *Polyarthra dolichoptera* * 23% | *Polyarthra dolichoptera* * 6% | *Polyarthra dolichoptera* * 35% |
| | *Bosmina longirostris* ** 17% | *Bosmina longirostris* ** 34% | *Bosmina longirostris* ** 7% | *Bosmina longirostris* ** 17% | *Bosmina longirostris* ** 8% | *Polyarthra remata* * 12% |
| | nauplius ** 7% | nauplius ** 6% | nauplius ** 5% | nauplius ** 13% | nauplius ** 12% | nauplius ** 4% |

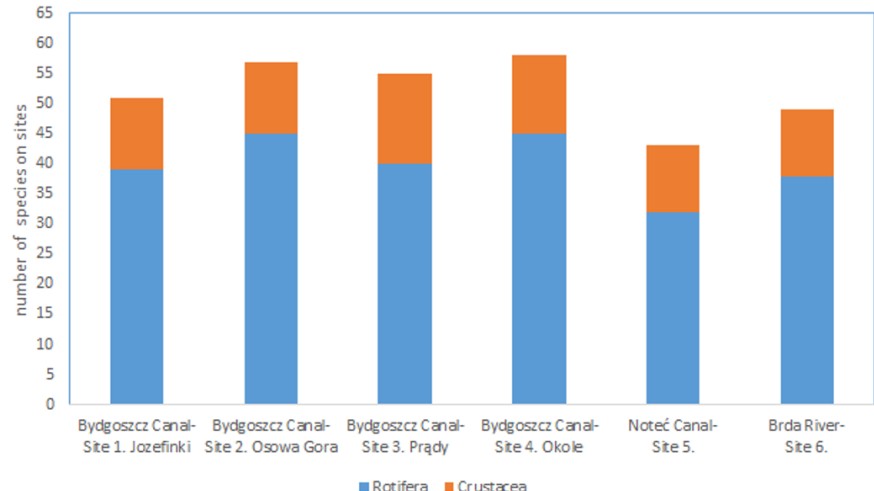

**Figure 2.** Number of zooplankton species during vegetation season in the Bydgoszcz Canal, the Noteć Canal, and the Brda River sites.

The average zooplankton density was 320 ind/L. Average zooplankton density was highest in the Bydgoszcz Canal (578 ind/L) and lowest in the Noteć Canal (196 ind/L) (Table 3). Average rotifer density was more than twice as high in the Bydgoszcz Canal (437 ind/L) as in the Brda River and more than three times as high as in the Noteć Canal. Average crustacean density was highest in the Bydgoszcz Canal (143 ind/L) and was significantly higher compared to the Noteć Canal (4×) or the Brda River (8×) (Table 3). Average zooplankton biomass was 3.026 mg/L, including 0.157 mg/L rotifer biomass and 2.869 mg/L crustacean biomass. The highest average zooplankton biomass was recorded in the Bydgoszcz Canal (4.258 mg/L), and the lowest in the Brda River (0.494 mg/L) (Table 3). Average crustacean biomass was more than seven times as high in the Bydgoszcz Canal (4.078 mg/L) as in the Noteć Canal (0.568 mg/L) and more than 12 times as high as in the Brda River (0.333 mg/L). During the vegetation season, the $\alpha$-diversity index (H' = 1.94 ± 0.03) was highest in the Noteć Canal, and evenness index was highest (J' = 0.71 ± 0.18) in the Bydgoszcz Canal, at site 4. The lowest $\alpha$-diversity index (H' = 1.85 ± 0.06) and evenness index (J' = 0.32 ± 0.21) were recorded in the Brda River (Table 3).

**Table 3.** Diversity (H′ index), evenness (J′ index), number of zooplankton species, zooplankton density, and biomass during vegetation season in the Bydgoszcz Canal, the Noteć Canal, and Brda River sites. Shannon–Weaver $\alpha$-diversity index (H′ index), Pielou's evenness index (J′ index), Tax total: number of species, Tax Rot: number of rotifers species, Tax Crust: number of crustacean species, N total: density of zooplankton (ind/L), N Rot: density of rotifers, N Crust: density of crustaceans, B total: biomass of species (µg/L), B Rot: biomass of rotifers species, and B Crust: biomass of crustacean species.

| | Bydgoszcz Canal | | | | | Noteć Canal | | Brda River | |
| | Site 1 | Site 2 | Site 3 | Site 4 | Mean-Range | Site 5 Mean | Range | Site 6 Mean | Range |
| --- | --- | --- | --- | --- | --- | --- | --- | --- | --- |
| Tax total | 16 | 17 | 17 | 21 | 18 (11–18) | 15 | (13–18) | 14 | (8–29) |
| Tax Rot | 11 | 13 | 13 | 16 | 13 (5–20) | 11 | (6–15) | 11 | (6–25) |
| Tax Crust | 5 | 4 | 4 | 5 | 4.5 (1–8) | 4 | (2–7) | 3 | (2–5) |
| N total | 574 | 332 | 662 | 744 | 578 (28–2478) | 196 | (25–640) | 246 | (16–1354) |
| N Rot | 429 | 247 | 577 | 496 | 437 (8–3430) | 161 | (15–620) | 229 | (14–1338) |
| N Crust | 150 | 88 | 86 | 249 | 143 (4–1420) | 35 | (10–78) | 17 | (2–38) |
| B total | 5.204 | 1.511 | 5.773 | 4.541 | 4.258 (0.113–26.285) | 0.634 | (0.108–1.661) | 0.494 | (0.026–1.349) |
| B Rot | 0.142 | 0.120 | 0.273 | 0.183 | 0.180 (0.004–0.971) | 0.066 | (0.003–0.243) | 0.161 | (0.027–0.882) |
| B Crust | 5.062 | 1.391 | 5.500 | 4.358 | 4.078 (0.046–26.259) | 0.568 | (0.099–1.656) | 0.333 | (0.023–0.383) |
| H′ index | 1.91 | 1.90 | 1.93 | 1.92 | 1.92 | 1.94 | | 1.85 | |
| J′ index | 0.57 | 0.50 | 0.49 | 0.71 | 0.57 | 0.48 | | 0.32 | |

### 3.2. Influence of Environmental Factors on Zooplankton Communities

The CCA revealed a relationship between the species composition and environmental variables at sites in the Bydgoszcz Canal, Noteć Canal, and Brda River. The distribution of environmental variables (vectors) along the axis clearly indicated that the significance of the variables depended on their length. Statistical significances were confirmed by results of the Spearman's rho test. Correlations and CCA were both calculated based on the original dataset (Figures 3–5).

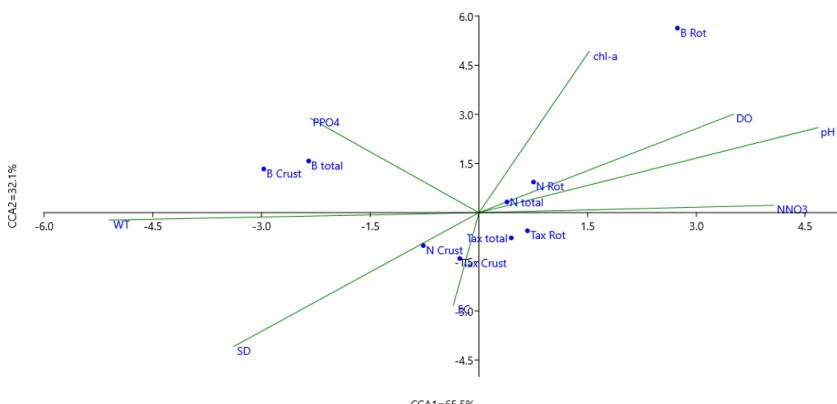

**Figure 3.** Results of canonical correspondence analysis (CCA) performed on zooplankton and environmental data during vegetation season at the Bydgoszcz Canal using forward selection of variables ($p < 0.01$). Triplot of significant environmental variables (water temperature: WT, Secchi disk visibility: SD, conductivity: EC, oxygen concentration: DO, pH, chlorophyll: chl-*a*, nitrate: $NNO_3^-$, phosphate: $PPO_4^{2-}$), number of zooplankton species (Tax total, Tax Rot, Tax Crust), zooplankton density (N total, N Rot, N Crust) and zooplankton biomass (B total, B Rot, B Crust).

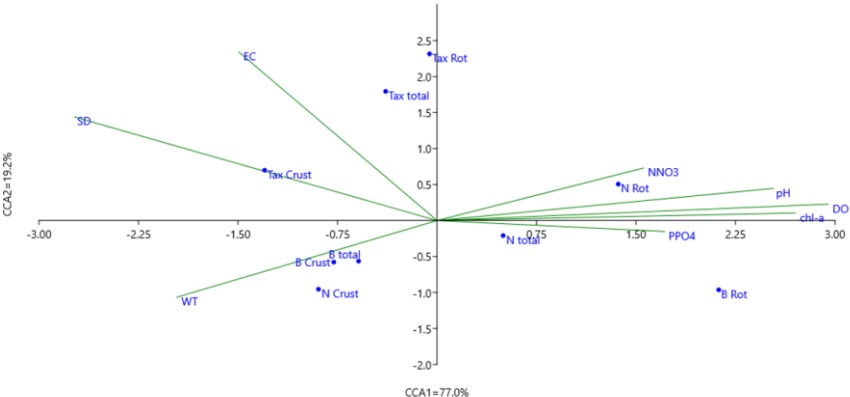

**Figure 4.** Results of canonical correspondence analysis (CCA) performed on zooplankton and environmental data during vegetation season in the Noteć Canal using forward selection of variables (*p* < 0.05). Triplot of significant environmental variables (water temperature: WT, Secchi disk visibility: SD, conductivity: EC, oxygen concentration: DO, pH, chlorophyll: chl-*a*, nitrate: $NNO_3^-$, phosphate: $PPO_4^{2-}$), number of zooplankton species (Tax total, Tax Rot, Tax Crust), zooplankton density (N total, N Rot, N Crust), and zooplankton biomass (B total, B Rot, B Crust).

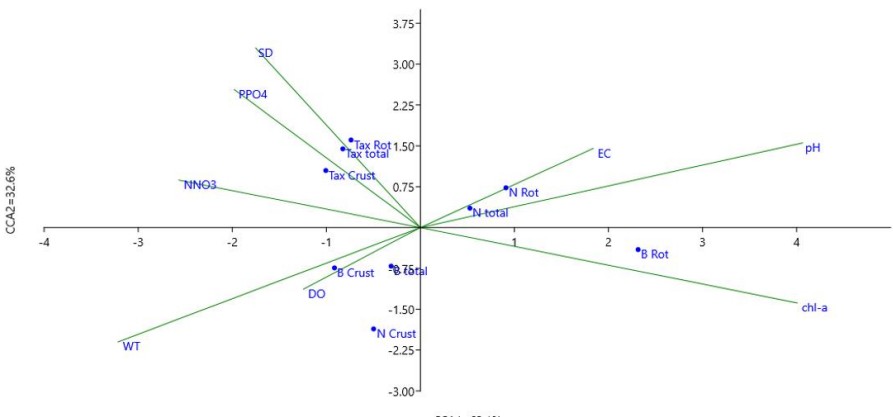

**Figure 5.** Results of canonical correspondence analysis (CCA) performed on zooplankton and environmental data during vegetation season in the Brda River using forward selection of variables (*p* < 0.05). Triplot of significant environmental variables (water temperature: WT, Secchi disk visibility: SD, conductivity: EC, oxygen concentration: DO, pH, chlorophyll: chl-*a*, nitrate: $NNO_3^-$, phosphate: $PPO_4^{2-}$), number of zooplankton species (Tax total, Tax Rot, Tax Crust), zooplankton density (N total, N Rot, N Crust), and zooplankton biomass (B total, B Rot, B Crust).

The triplot at sites of the Bydgoszcz Canal showed that the eigenvalues of the first ($\lambda$CCA1 = 0.655) and second ($\lambda$CCA2 = 0.321) CCA axes accounted for 97.6% of the variation in the environmental data (Figure 3). The distribution of environmental variables along the axis indicated the following significances related to primary production: changes in water pH and nitrate concentration showed negative correlation with water temperature (r = −0.801; *p* ≤ 0.01 and r = −0.664; *p* ≤ 0.01). Conversely, water pH had a positive relationship with dissolved oxygen (r = 0.735, *p* ≤ 0.01). Chlorophyll concentration was negatively correlated with Secchi disk visibility (r = −0.783; *p* ≤ 0.01). These relationships result largely from the distribution of these variables in relation to the first axis, which describes 65% of the total variability used for analyses. According to CCA, changes in the biomass and density of rotifers showed the greatest dependence on chlorophyll concentration (r = 0.792, *p* ≤ 0.01 and r = 0.748, *p* ≤ 0.01). On the other hand, the biomass of rotifers and their density changed inversely to Secchi disk visibility (r = −0.642, *p* ≤ 0.01) and (r = −0.679, *p* ≤ 0.01). The biomass, density, and species number of crustacean were most correlated with changes in water temperature (r = 0.692, *p* ≤ 0.01; r = 0.857, *p* ≤ 0.01;

r = 0.725, $p \leq 0.01$). Crustacean density also showed a negative correlation with changes in pH and nitrate concentration (r = $-0.744$, $p \leq 0.01$ and r = $-0.685$, $p \leq 0.01$) (Figure 3).

The triplot at the Noteć Canal site showed that the eigenvalues of the first CCA axis ($\lambda$CCA1 = 0.770) and the second ($\lambda$CCA2 = 0.192) accounted for 96.2% of the variation in the environmental data (Figure 4). The variability of data identified by the second axis was slightly different, and much higher than in the Bydgoszcz Canal. This resulted from the variability of WT, SD, and EC. The distribution of environmental variables along the axis indicated the following significances related to primary production: changes in water pH and nitrate concentration showed a negative trend with water temperature (r = $-0.857$, $p \leq 0.05$ and r = $-0.929$, $p \leq 0.05$). The CCA indicated the existence of trends similar to those in the Bydgoszcz Canal. Only the relationship between chlorophyll concentration and density of rotifers (r = 0.757, $p \leq 0.05$), as well as total density of zooplankton (r= 0.815, $p \leq 0.05$), turned out to be statistically significant (Figure 4).

The triplot at the Brda River site showed that the eigenvalues of the first CCA axis ($\lambda$CCA1 = 0.631) and the second ($\lambda$CCA2 = 0.326) accounted for 95.7% of the variation in the environmental data (Figure 5). The variability of the analyzed data explained the difference in distribution compared to the previously analyzed canals. The CCA showed that the higher concentration of dissolved oxygen did not indicate changes in water pH. On the other hand, concentration of dissolved oxygen was negatively correlated with Secchi disk visibility (r = $-0.896$, $p \leq 0.05$). The analyses showed that there was no relationship between nutrient content and primary algae production. Only rotifers abundance showed a positive trend with chlorophyll concentration (r = 0.847, $p \leq 0.05$).

Two-way cluster analysis was used to select between environmental parameters at the studied sites during the growing season (Figure 6A). The dendrogram showed a good division between sites based on water temperature, pH, and conductivity. Within the studied sites and months, the cluster analysis characterized a high correlation between the Noteć Canal, the Bydgoszcz Canal, and the Brda River in spring months. The cluster analysis showed differences between the Brda River and both studied canals in summer and autumn. Meanwhile, at the same time (summer and autumn), the correlation between the Bydgoszcz Canal and the Noteć Canal was observed (Figure 6A). Two-way cluster analysis also compared the selected biological parameters (zooplankton data) at the studied sites during the growing season (Figure 6B). The dendrogram showed a good division between sites within individual months based on average zooplankton density, rotifers density, and biomass. The basic division distinguished two groups. The first group comprised autumn and summer samples primarily from the Noteć Canal and Brda River, and the second group comprised samples from the Bydgoszcz Canal sites prevailing during spring, summer, and autumn. This division indicated the greatest differentiation of zooplankton in the Bydgoszcz Canal, e.g., the largest number of samples (September and August) was outside the groups consisting mainly of samples from spring, summer, and autumn (Figure 6B).

Two-way cluster analysis compared the environmental parameters and group of sites (Figure 7A). The dendrogram showed a good division between average water temperature and pH, as well as between Secchi disk visibility, conductivity, and chlorophyll. In terms of environmental parameters, the Brda River was significantly different from the water in canal sites. The parameters at site 4 were also different from those found at the other sites of the Bydgoszcz Canal. We used two-way cluster analysis to compare the biological parameters (zooplankton data) at the studied sites (Figure 7B). The dendrogram showed a good division between number of zooplankton species, total zooplankton density, and biomass. The cluster analysis divided sites into two groups: the first group comprised the Bydgoszcz Canal (sites 1, 3, and 4), and the second group comprised the Brda River (site 6) with the Noteć Canal and site 2 from the Bydgoszcz Canal (site 5) (Figure 7B).

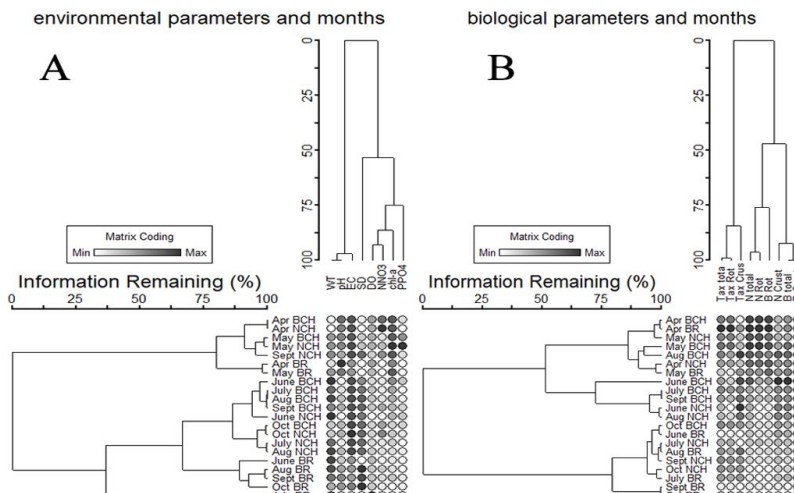

**Figure 6.** Tree diagrams of cluster analysis of study sites divided for seven months within vegetation season based on environmental parameters (**A**) and biological parameters (**B**) (zooplankton data) obtained using Ward's method as linkage rule and Euclidean distances as the metric for distance calculation. BCH: Bydgoszcz Canal, NCH: Noteć Canal, BR: Brda River.

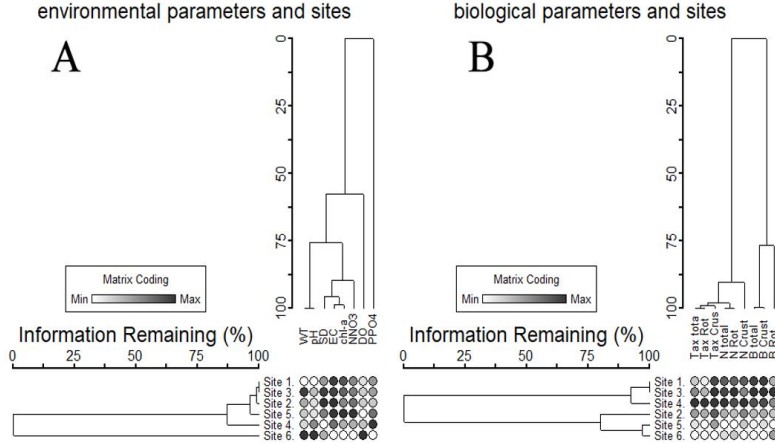

**Figure 7.** Tree diagrams of cluster analysis of study sites based on environmental parameters (**A**) and biological parameters (**B**) (zooplankton data) obtained using Ward's method as linkage rule and Euclidean distances as the metric for distance calculation.

The results from statistical analyses showed good division between the sites based on environmental and biological parameters. In the Noteć Canal and Brda River, Secchi disk visibility, chlorophyll concentration, and dissolved oxygen were driving zooplankton variation. The results from sites in Bydgoszcz Canal showed that water temperature, dissolved oxygen concentration of nitrate, and chlorophyll were the dominant parameters shaping the zooplankton variation.

## 4. Discussion

During the study, we identified the most common and cosmopolitan zooplankton species, including the rotifer species *Keratella cochlearis*, *Keratella quadrata*, and *Polyarthra dolichoptera*, and the crustacean species Cladocera *Bosmina longirostris* and nauplii (copepod larval forms). Napiórkowski and Napiórkowska [26] studied zooplankton in the small, free-flowing Wel River, where the zooplankton species composition was comparable with our results. Rotifers dominated over crustaceans, which was caused by their better adaptation to adverse environmental conditions of lotic and semi-lotic habitats [43,44]. Rotifers dominated in species number and density in all studied watercourses, as in the Illinois

river [45]. An increase was observed in the share of their density and biomass in the total biomass of spring zooplankton. This was influenced by the lower temperatures recorded in spring and the higher chlorophyll concentration (turbid period [46]). The small algae that appear in spring provide excellent food for rotifers, and this favors the development of rotifer zooplankton, that is to say, lower temperatures and chlorophyll correlated with rotifer abundance and biomass. Our results showed that *Keratella cochlearis* was the most abundant in the canals, and *Polyarthra dolichoptera* in the river (Table 2). Numerous researchers have observed that these taxa tend to prevail in river systems [47–49]. The zooplankton density increased mainly because small rotifers tolerate water flow variability. Many authors have noted that a high density of rotifers in both standing and flowing waters results from their tolerance to diverse environmental conditions [8,50–52]. Rotifers' dominance is related to their small size and rather short development time in comparison to crustaceans [53–55].

In our study, active filter-feeding crustaceans (*Daphnia magna*, *Chydorus sphaericus*, and *Diaphanosoma brachyurum* and larval forms of copepods (assumed to be *Acanthocyclops*, based on its predominance among mature copepods in samples)) were rarely represented. The results of our studies indicated lower crustacean density. This zooplankton group could be affected by the following factors: (1) unfavorable hydrological conditions (increased water velocity, turbulent water flow), (2) unfavorable nutritional conditions (low food availability may be connected with lower trophic status), (3) fish predation pressure, and (4) lack of macrophytes (which normally offer shelter or refuge for large crustaceans). The number of taxa, density, and biomass of crustaceans decreased in the gradient of flows recorded at sites from the Bydgoszcz Canal, through the Noteć Canal to the Brda River. The very slow flow on the Bydgoszcz Canal, caused by hydrotechnical constructions, favored the development of macrophytes as ecological niches for crustacean zooplankton [56,57]. There were four times more crustaceans in the Bydgoszcz Canal than in the Noteć Canal and over eight times more than in the Brda River. In stagnant water retained by sluices or dams, zooplankton is more abundant than in the river itself [7,58]. Due to water stagnation, the zooplankton of floodplains is also richer than the zooplankton of river channels [59,60]. In another study, Czerniawski and Kowalska-Góralska [61] investigated zooplankton in free-flowing lotic waters with small dams. Presumably, the dams broke the continuity of the river and affected zooplankton distribution by causing rapid hydrological changes (reduction in current velocity, increase in water retention time, increase in water temperature, and increase in nutrient content) [62–65]. Similar conditions were observed in the studied canal, where the continuity was interrupted by hydrotechnical structures, e.g., locks.

In our study, crustacean species contributed up to 95% of the total zooplankton biomass. Crustacean biomass was highest in the Bydgoszcz Canal and lowest in the Brda River, while rotifer biomass was comparable in all the waterways. The crustacean zooplankton biomass increased rapidly in summer, when the rotifer biomass decreased (due to the impact of temperature and macrophytes development) [57]. The biomass of crustaceans was most correlated with changes in water temperature (Figure 3). Macrophytes formed an excellent refugium for zooplankton development [56,66]. This influenced results throughout the study period. In spring, the canal waters were dominated by algae, which contributed to a decrease in transparency. The change from turbid to clear water in summer was reminiscent of the alternative stable states in lakes studied by Scheffer and Jeppesen [46]. There were far fewer macrophytes in the waters of the Noteć Canal, even during summer. This was connected with flow being 0.25 m/s faster than in the Bydgoszcz Canal. The main current of the Noteć Canal was divided into two parts. One of them was constantly flowing through the turbines of small hydroelectric power plants, while the other was slightly slowing in front of the locks. The Brda River was characterized by higher water flow (0.8 m/s) compared to the canals, which was not conducive to the development of macrophytes [67,68]. A fast water current is a significant obstacle to the development of zooplankton.

Unfortunately, there are not many studies focusing on zooplankton species composition in canals. Several studies have used a similar approach to investigate zooplankton in small rivers with low flow velocity and in stagnant waters [7,9,26]. However, zooplankton production in stagnant and slow-flowing rivers is very important because these water bodies are a major source of zooplankton in river–lake systems [59,69,70].

The CCA showed a similar relationship between the environmental conditions and the structure of zooplankton found in Bydgoszcz Canal and the Noteć Canal (Figures 3 and 4). The CCA indicated the significance of the variables related to primary production, i.e., oxygen concentration, water pH, and chlorophyll concentration, for which water transparency and its temperature were negatively correlated (Figure 3). The diagram for the Noteć Canal indicated the existence of similar trends as in the Bydgoszcz Canal (Figure 4). In the case of the Brda River, CCA indicated a relationship between pH and oxygen concentration that was different than in the previous cases (higher oxygen content in the samples was not recorded at the highest pH values) (Figure 5). The analysis of the Brda also showed that there was no relationship between changes in SD and DO and between nutrient content and primary algae production (Figure 5). Hence, the zooplankton habitat in the river is significantly different than in the canals, possibly due to specific hydrological conditions. During the growing season in the Bydgoszcz Canal, the abundance and biomass of rotifers changed similarly to chlorophyll (Figure 3). By contrast, there is no recorded correlation between chlorophyll and crustaceans. It is likely that small phytoplankton (chlorophyll) appearing in spring provide excellent food for rotifers [43,71]. Shayestehfar et al. [72] emphasize that rotifer density and distribution depend on the variety of ecological and physicochemical factors such as food availability, but also temperature, water pH, and their relationships with other organisms. All these factors play an important role in determining variations in rotifer density [73]. The main factor affecting the abundance and biomass of crustacean zooplankton, as well as the total zooplankton in the Bydgoszcz Canal, was water temperature. A similar relationship was observed on the Danube [74]. This relationship was observed in the CCA, and its statistical significance was confirmed by Spearman rank R coefficient.

The total zooplankton biomass was strongly correlated with water temperature in the Bydgoszcz Canal (Figure 3). Similar results were reported by Hansson et al. [75], suggesting that the spring period, with strong alterations in temperature-driven processes such as predation and resource supply, is important in shaping the summer zooplankton community. For example, moderate temperatures in May accelerated the growth and feeding rate of many small feeders (rotifers) [76]. In the Noteć Canal, the density of rotifers was significantly correlated with chlorophyll, and, in the Brda River, rotifer biomass was also significantly correlated with chlorophyll (Figure 5). It is likely that, in faster flowing watercourses, fine planktonic diatoms and coccal chlorophyta may be an important part of the rotifers' diet [71].

The results of two-way cluster analysis highlighted the differentiation of environmental and biological conditions between habitats in canals and in the river (Figure 6A,B; Figure 7A,B). According to environmental conditions, the sites in the Bydgoszcz and Noteć Canals were separated from the Brda River (Figure 7A). According to biological parameters, the results indicated similar zooplankton structures at the sites on the Bydgoszcz Canal; only site 2 was more similar to the Noteć Canal and the Brda River (Figure 7B). The similarity could be explained by the fact that site 2 is located near the Noteć Canal's mouth to the Bydgoszcz Canal.

The locations in the Bydgoszcz Canal were characterized by high numbers of zooplankton species, indicating an optimal range of environmental variables. Some authors suggest that zooplankton communities in rivers depend largely on environmental conditions: their low variability enhances zooplankton growth [17,77,78].

Differences between the Bydgoszcz Canal, Noteć Canal, and Brda River occurred throughout the growing season and were probably due to the different hydrological conditions (water flow variability) prevailing at the study sites and the different levels of

macrophyte vegetation. Water flow may directly influence the environmental conditions and the development of zooplankton organisms [26,79,80], or indirectly by allowing macrophytes to create ecological niches supporting zooplankton development [56,81]. Both direct and indirect effects of hydrological conditions on zooplankton life were observed in the studied watercourses.

## 5. Conclusions

During our studies the significance of the variables related to primary production, i.e., oxygen concentration, water pH, and chlorophyll concentration, was observed in the Bydgoszcz Canal and Noteć Canal. The primary production variables shaped the zooplankton community, especially density and biomass of rotifers in the studied canals.

The Bydgoszcz Canal is richer in zooplankton (density, biomass, and number of species) compared to the Brda River or the Noteć Canal. The reason may be different hydrological conditions, e.g., slower water flow (in Bydgoszcz Canal) directly influencing zooplankton development by creating more stable growth conditions. Locks on the Bydgoszcz Canal reduce water flow. This had an indirect influence by increasing the number of macrophytes that create ecological niches, in turn benefitting the development of zooplankton organisms, especially crustaceans.

The results of two-way cluster analysis according to environmental conditions showed that the sites in the Bydgoszcz Canal and the Noteć Canal were separated from those in the Brda River. Whereas, according to the biological parameters, the results indicated similarity of zooplankton structure among sites 1, 3, and 4 of the Bydgoszcz Canal, only site 2 stood out, being more similar to the Noteć Canal and the Brda River.

The analysis of the Brda River showed that there was a lack of relationship between changes in SD and oxygen concentration and between nutrient content and primary algae production (chlorophyll). Hence, the zooplankton habitat in the river is significantly different from that found in the studied canals, possibly due to specific environmental conditions, e.g., hydrology.

**Author Contributions:** Conceptualization, P.N. and N.K.; methodology, P.N.; software, N.K.; validation, N.K. and P.N.; formal analysis, N.K. and P.N.; investigation, N.K. and P.N.; resources, N.K.; data curation, N.K.; writing—original draft preparation, N.K.; writing—review and editing, P.N.; visualization, N.K.; supervision, P.N.; project administration, P.N.; funding acquisition, P.N. All authors have read and agreed to the published version of the manuscript.

**Funding:** This study was supported by the Polish Minister of Science and Higher Education, under the program "Regional Initiative of Excellence" in 2019–2022 (grant No. 008/RID/2018/19).

**Institutional Review Board Statement:** Not applicable.

**Informed Consent Statement:** Not applicable.

**Data Availability Statement:** Data available on request due to restrictions, e.g., privacy or ethical. The data presented in this study are available on request from the corresponding author. The data are not publicly available due to preparing of PhD dissertation.

**Conflicts of Interest:** The authors declare no conflict of interest. The funders had no role in the design of the study; in the collection, analyses, or interpretation of data; in the writing of the manuscript, or in the decision to publish the results.

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
