# Peer review of "How Do Specific Environmental Conditions in Canals Affect the Structure and Variability of the Zooplankton Community?"

_water, doi:10.3390/w14060979_

Round 1

Reviewer 1 Report

There are seldom studies focusing on zooplankton species composition in canals The manuscript present comprehensive environmental monitoring and investigating the relationship between zooplankton. It is suggested to publish.

Author Response

Responses to the reviewer's comments - Reviewer #1

Thank You

Reviewer 2 Report

The manuscript is an attempt to define "how do specific environmental conditions [....] affect the structure and variability of the zooplankton community" (as the authors themselves put it in the title). The physical and chemical properties of the water that are responsible for the growth of algae were selected for the study, i.e. various forms of nutrients and water temperature, visibility of the Secchi disk, electrolytic conductivity, oxygen and chlorophyll content in the water of the canals and the river. The inclusion of the content of chlorophyll a in the group of physico-chemical properties of water raises concerns, as it characterizes the biomass of another ecological group - planktonic algae, which are food for the studied taxonomic groups of zooplankton. Therefore, it should be treated as a biological element.

On the basis of these properties, the authors attempted to characterize the diversity of the studied habitats for zooplankton, and thus to characterize the differences in their communities.

I consider the choice of research objects justified, i.e. giving a chance to formulate an answer to the question posed in the title.

Unfortunately, the manuscript, though carefully prepared at first glance, contains some common simplifications and requires changes. First, the "Material and methods" section requires extension, at least with information related to the canals and the river, which in the current version of the manuscript appear only in the discussion. I am thinking of the presence (or absence) of hydrotechnical structures and aquatic vegetation, which is indicated in the discussion as a refuge for zooplankton. Nevertheless, it can be clearly stated that the study did not carry out any analyzes related to the mentioned aim of "creating ecological niches for zooplankton" by submerged macrophytes (line 66). This issue appears only in the discussion, but the seasonal nature of plants is not mentioned at all, which would be expected based on the purpose of the study (lines 62-66).

Another drawback of the manuscript is the fact that the discussion of the results shows that the more important factors shaping the structure and abundance of zooplankton in slow-flowing rivers (i.e. lowland rivers) are those that are only mentioned in the discussion, i.e. flow velocity, presence or absence of macrophytes and the presence of hydrotechnical structures. The majority (at least 2/3) of the text was devoted to the discussion of such relationships, and the discussion of the studied relationships between the environmental properties selected for analyzes was actually only used to confirm the previously demonstrated relationships. This is clearly emphasized by the content of the section - conclusions - out of the four presented conclusions, only one (and the shortest one) directly relates to the research (statistical analyzes) described in the "Results" section.

Additionally, the reading of the text is made even more difficult by the lack of consistency in defining and recalling the names of research sites and the parameters described. For example, the water temperature is sometimes abbreviated as WT (line 120) to replace this abbreviation with Tw (Table 1) a few lines below, and to return to WT again in line 135, etc.

On the other hand, sampling stations are sometimes marked with a Polish name, in another place of the text - with a number (lines 94-98 and Fig. 1). However, mistakes can also be found here. For example, Osowa Gora receives the number 2 (site 2) in line 95 and figure 2, but in lines 159 and 165 it has the number 4 assigned to it, and in figure 7 it is presented only as 'Osowa'. Perhaps, however, in the English text it would be worth limiting ourselves to the numbers only (leaving their geographical names in the "Material and methods" section).

Detailed comments:

Lines 36-37: The last sentence of this paragraph would fit (with appropriate modifications) to lines 56-61.

Lines 55-56: I would delete this sentence, leaving quote [23] at the end of the previous sentence.

Lines 94-98: The entire paragraph should be placed after the sentence "The width of the Brda River ...." (line 107).

Lines 100-101: The description of the drawing shows that all sampling stations in the Bydgoszcz Canal were located at the sluices (which is not mentioned in the text at all), while the reader does not find information about the exact location of each of these points in relation to the hydrotechnical structure and the direction of the water flow (very weak indeed, but still). In front of the sluice, behind the sluice ?

Lines 127-130: If units are quoted in a table, it doesn't make sense to include them in the headline, line 129, "chl-a", in the table "Chl-a".

Line 131 (Table 1): How to understand the zero visibility of a Secchi disc? My guess is that the lush macrophytes prevented the measurement. However, the value of zero in this situation may not correspond to the actual light conditions in the water at the site.

Lines 139-142: The adopted method of preparing the data for CCA analysis essentially differs from the traditional method of conducting it, instead of the output data, only mean values from the study period were compared. The aforementioned problem of outliers was eliminated by the transformation, and calculations only on average values excluded (almost) all data variability, including the one resulting from the seasonal variation mentioned in the target.

Line 149-150: The applied transformation does not eliminate the problem of the presence of null values.

Line 167: Description of the bars showing the number of zooplankton in the Notec Canal and the Brda River in Figure 2 should be completed with the numbers of the stands.

Lines 191-195: Unify the symbols in the table and its header, respectively: H 'and “H' index”, Taxtotal and Taxtot, Ntotal and Ntot, Btotal and Btot.

Line 199: Figure 3 shows the effect of the CCA performed on the mean values for the four sites, not the analysis "at the studied sites" .

Line 215: The figure shows an ordering diagram, not a triplot.

Lines 203-211: I suggest changing the description of the CCA results. Firstly, the distribution of environmental variables along the axis clearly indicates the significance of the variables related to primary production, i.e. oxygen content, water pH and chlorophyll concentration,

for which water transparency and its temperature were negatively correlated. These relationships result primarily from the distribution of these variables in relation to the first axis, which describes as much as 83% of the total variability of the data used for the analysis! Secondly, according to the CCA analysis, the biomass of rotifers showed the greatest dependence of changes with the concentration of chlorophyll, and their number, in turn, with the content of dissolved oxygen and water pH. On the other hand, the values ​​of biomass (especially) and the number of rotifers changed contrary to the visibility of the Secchi disc. The number and biomass of crustaceans were most correlated with changes in water temperature, and on the contrary with changes in NNO3. The main environmental gradient, already described above, had no effect on the number of zooplankton taxa at all, on the number of rotifers taxa to a very small extent, and on the number of crustacean species somewhat more. These variables changed the most along the gradient described by the second axis of the diagram, i.e. in line with 12.5% ​​of the overall variability of the analyzed environmental parameters. The statistical significance of the trends described is confirmed (or not) by the results of correlation analyzes using the Spirman test described in lines 205-211. They are slightly different, but are these differences not the result of using other data sets (means were used for CCA, but the correlations were probably calculated on the original data)?

Lines 216-219: There are NNO3 and PPO4 in the chart.

Lines 220-222: The ordering diagram clearly shows that in the Notec Canal the variability of data identified by the second axis was slightly different - much higher than in the Bydgoszcz Canal. It resulted from the variability of WT, SD and EC. However, due to the lack of statistical significance of the trends indicated by the CCA in the description of this figure, I would avoid the expressions "was correlated", and just emphasized that the diagram indicated the existence of similar trends as in the Bydgoszcz channel. Only the relationship between Chl-a and the density of rotifers, and Chl-a and total zooplankton turned out to be statistically significant.

Lines 237-243: In this case, apart from the already mentioned different distribution of the explainable variability contained in the analyzed data (63% and 33%) than the previously analyzed channels, it is surprising to indicate different than in the previous cases (basically in nature) relationships between pH and oxygen content. The diagram shows that the higher oxygen content in the samples was not recorded at (highest) pH values, which may indicate that the river water was contaminated with substances that increase the water pH. The analysis also shows that there is no relationship between changes in SD and DO, and between nutrient content and primary algae production. Hence, the zooplankton habitat in the river is significantly different than in the canals (assessment based on the analyzed variables).

Lines 265-269: Of course, chlorophyll-a does not describe the physico-chemical properties.

Lines 161-264: According to the horizontally oriented dendrogram in Fig. 6B, the basic division distinguished two groups, the "lower" group comprising rather samples from autumn (Oct - all studied habitats, i.e. BR, NCH and BCH, Sept - BR and NCN) and some summer habitats BR (June, Aug) and NCH (July ). The second group is the spring rehearsals - 6 rehearsals from May and April + Aug from BCH, and the summer rehearsals from both channels. In my opinion, this division indicates the greatest differentiation of zooplankton in the Bydgoszcz Canal - the largest number of samples (Sept and Aug) was outside the groups consisting mainly of samples from spring, summer and autumn.

Lines 280-281: I consider both triplots to be the most important at work, but the question arises whether all the graphs previously posted are actually necessary? Fig. 7a shows that in terms of physicochemical properties and chlorophyll content, the water in Brda is significantly different from the water in all canals, but also that the properties in the Okole site are significantly different from those found at other sites in the Bydgoszcz canal . (You can already see it in Table 1.) However, a question arises - was it correct to carry out the calculations on the average values calculated for all positions in the Bydgoszcz Canal? In my opinion, the answer is unequivocal - NO (especially at the stage of CCA analyzes). It is surprising, however, that it is not the zooplankton at the Okole site on the 7B triplot that is more similar to the one from the Brda and the Notec Channel. It turns out that the zooplankton in the Osowa site appears to be the most similar to the zooplankton in the Brda River and the Notec Channel.

Lines 304-308 and 313-319: Much of this information is not found in the Materials and Methods section, and therefore where it should be.

Lines 376-378: The ordering diagram does not show that rotifers abundance and biomass correlate with chlorophyll and DO only in spring - it shows that they changed similarly to temperature and DO.

Lines 404-405: Evidently untrue - the zooplankton at the Osowa site was more similar to that of the Brda River and the Notecki Canal than other sites in the Bydgoszcz Canal!

Author Response

The authors would like to thank the Scientific Editor for the constructive comments and for the insightful review of second and third Reviewer. We tried to improve the  first CCA figure, results and discussion chapters and we reformulate conclusions. The results and discussion chapters have been partially rewritten. We add information about water flow to bold difference between canals and river. According to Editor suggestions we have improved all tables, and citations of tables in text of MS. We use the professional English proofing service.

Responses to the reviewer's comments - Reviewer #2

  1. The manuscript is an attempt to define "how do specific environmental conditions [....] affect the structure and variability of the zooplankton community" (as the authors themselves put it in the title). The physical and chemical properties of the water that are responsible for the growth of algae were selected for the study, i.e. various forms of nutrients and water temperature, visibility of the Secchi disk, electrolytic conductivity, oxygen and chlorophyll content in the water of the canals and the river. The inclusion of the content of chlorophyll a in the group of physico-chemical properties of water raises concerns, as it characterizes the biomass of another ecological group - planktonic algae, which are food for the studied taxonomic groups of zooplankton. Therefore, it should be treated as a biological element.

Chlorophyll as very important water quality parameter was added into the group of environmental variables exchanged for previous physico-chemical parameters, biological elements represented zooplankton group. 

  1. Unfortunately, the manuscript, though carefully prepared at first glance, contains some common simplifications and requires changes. First, the "Material and methods" section requires extension, at least with information related to the canals and the river, which in the current version of the manuscript appear only in the discussion. I am thinking of the presence (or absence) of hydrotechnical structures and aquatic vegetation, which is indicated in the discussion as a refuge for zooplankton. Nevertheless, it can be clearly stated that the study did not carry out any analyzes related to the mentioned aim of "creating ecological niches for zooplankton" by submerged macrophytes (line 66). This issue appears only in the discussion, but the seasonal nature of plants is not mentioned at all, which would be expected based on the purpose of the study (lines 62-66).

Information about hydrotechnical structures (sluices) on the Bydgoszcz canal is added in the materials and methods chapter. This section also describes the occurrence of macrophytes species which developed intensively in the summer months near of hydrotechnical structures (sluices).

  1. Another drawback of the manuscript is the fact that the discussion of the results shows that the more important factors shaping the structure and abundance of zooplankton in slow-flowing rivers (i.e. lowland rivers) are those that are only mentioned in the discussion, i.e. flow velocity, presence or absence of macrophytes and the presence of hydrotechnical structures. The majority (at least 2/3) of the text was devoted to the discussion of such relationships, and the discussion of the studied relationships between the environmental properties selected for analyzes was actually only used to confirm the previously demonstrated relationships. This is clearly emphasized by the content of the section - conclusions - out of the four presented conclusions, only one (and the shortest one) directly relates to the research (statistical analyzes) described in the "Results" section.

Discussion chapter was rebuilt; the first part is devoted to the identification of zooplankton species representing the studied area in relation to the measured environmental parameters. The second part describes the diversity and abundance of zooplankton in the context of the hydrological situation - the rate of water flow in the canals and in the river. The third part points to the presence of macrophytes, which began to develop when the flow of water slowed down. Macrophytes created suitable hatching conditions for the development of crustacean zooplankton species. The forth part is devoted to statistical analysis, which concerns the correlation of relationships between environmental and biological parameters. Hence, we were interested which environmental parameters affect the biological parameters at sites depending of growing season. The last part is devoted to discussion summary. Conclusions were written based on these findings.

  1. Additionally, the reading of the text is made even more difficult by the lack of consistency in defining and recalling the names of research sites and the parameters described. For example, the water temperature is sometimes abbreviated as WT (line 120) to replace this abbreviation with Tw (Table 1) a few lines below, and to return to WT again in line 135, etc.

We corrected this abbreviation.

  1. On the other hand, sampling stations are sometimes marked with a Polish name, in another place of the text - with a number (lines 94-98 and Fig. 1). However, mistakes can also be found here. For example, Osowa Gora receives the number 2 (site 2) in line 95 and figure 2, but in lines 159 and 165 it has the number 4 assigned to it, and in figure 7 it is presented only as 'Osowa'. Perhaps, however, in the English text it would be worth limiting ourselves to the numbers only (leaving their geographical names in the "Material and methods" section).

The names of sites were unified and numbered. The original Polish names remained in the material and method.

  1. Lines 36-37: The last sentence of this paragraph would fit (with appropriate modifications) to lines 56-61.

We followed reviewer suggestion.

  1. I would delete this sentence, leaving quote [23] at the end of the previous sentence.

It was deleted.

  1. Lines 94-98: The entire paragraph should be placed after the sentence "The width of the Brda River ...." (line 107).

It was removed.

  1. Lines 100-101: The description of the drawing shows that all sampling stations in the Bydgoszcz Canal were located at the sluices (which is not mentioned in the text at all), while the reader does not find information about the exact location of each of these points in relation to the hydrotechnical structure and the direction of the water flow (very weak indeed, but still). In front of the sluice, behind the sluice ?

We added detailed description about sites locations.

  1. Lines 127-130: If units are quoted in a table, it doesn't make sense to include them in the headline, line 129, "chl-a", in the table "Chl-a".

We changed it.

  1. Line 131 (Table 1): How to understand the zero visibility of a Secchi disc? My guess is that the lush macrophytes prevented the measurement. However, the value of zero in this situation may not correspond to the actual light conditions in the water at the site.

We corrected values of Secchi disk visibility, instead of zero value (it was mistake) in table 1 is written minimum measured value.

  1. Lines 139-142: The adopted method of preparing the data for CCA analysis essentially differs from the traditional method of conducting it, instead of the output data, only mean values from the study period were compared. The aforementioned problem of outliers was eliminated by the transformation, and calculations only on average values excluded (almost) all data variability, including the one resulting from the seasonal variation mentioned in the target.

In statistical analysis were used and compared real “raw” data from study period.  

  1. Line 149-150: The applied transformation does not eliminate the problem of the presence of null values.

We corrected it.

  1. Line 167: Description of the bars showing the number of zooplankton in the Notec Canal and the Brda River in Figure 2 should be completed with the numbers of the stands.

We improved it.

  1. Lines 191-195: Unify the symbols in the table and its header, respectively: H 'and “H' index”, Taxtotal and Taxtot, Ntotaland Ntot, Btotal and Btot.

We unified it.

  1. Line 199: Figure 3 shows the effect of the CCA performed on the mean values for the four sites, not the analysis "at the studied sites" .

We completely rebuilt this paragraph. The CCA is performed on the real dataset.

  1. Line 215: The figure shows an ordering diagram, not a triplot.

Presented CCA figures are triplots.

  1. Lines 203-211: I suggest changing the description of the CCA results. Firstly, the distribution of environmental variables along the axis clearly indicates the significance of the variables related to primary production, i.e. oxygen content, water pH and chlorophyll concentration, for which water transparency and its temperature were negatively correlated. These relationships result primarily from the distribution of these variables in relation to the first axis, which describes as much as 83% of the total variability of the data used for the analysis! Secondly, according to the CCA analysis, the biomass of rotifers showed the greatest dependence of changes with the concentration of chlorophyll, and their number, in turn, with the content of dissolved oxygen and water pH. On the other hand, the values ​​of biomass (especially) and the number of rotifers changed contrary to the visibility of the Secchi disc. The number and biomass of crustaceans were most correlated with changes in water temperature, and on the contrary with changes in NNO3. The main environmental gradient, already described above, had no effect on the number of zooplankton taxa at all, on the number of rotifers taxa to a very small extent, and on the number of crustacean species somewhat more. These variables changed the most along the gradient described by the second axis of the diagram, i.e. in line with 12.5% ​​of the overall variability of the analyzed environmental parameters. The statistical significance of the trends described is confirmed (or not) by the results of correlation analyzes using the Spirman test described in lines 205-211. They are slightly different, but are these differences not the result of using other data sets (means were used for CCA, but the correlations were probably calculated on the original data)?

We changed the Bydgoszcz Canal figure and description belonging to CCA analyses according reviewer suggestion. Data for CCA analyses as well as data for Spearman correlations were calculated on the original data set. 

  1. Lines 216-219: There are NNOand PPO4in the chart.

 Unfortunately, in this statistic analyze is not possible to use Subscript or Superscript.

  1. Lines 220-222: The ordering diagram clearly shows that in the Notec Canal the variability of data identified by the second axis was slightly different - much higher than in the Bydgoszcz Canal. It resulted from the variability of WT, SD and EC. However, due to the lack of statistical significance of the trends indicated by the CCA in the description of this figure, I would avoid the expressions "was correlated", and just emphasized that the diagram indicated the existence of similar trends as in the Bydgoszcz channel. Only the relationship between Chl-a and the density of rotifers, and Chl-a and total zooplankton turned out to be statistically significant

We followed reviewer suggestion and changed this paragraph.

  1. Lines 237-243: In this case, apart from the already mentioned different distribution of the explainable variability contained in the analyzed data (63% and 33%) than the previously analyzed channels, it is surprising to indicate different than in the previous cases (basically in nature) relationships between pH and oxygen content. The diagram shows that the higher oxygen content in the samples was not recorded at (highest) pH values, which may indicate that the river water was contaminated with substances that increase the water pH. The analysis also shows that there is no relationship between changes in SD and DO, and between nutrient content and primary algae production. Hence, the zooplankton habitat in the river is significantly different than in the canals (assessment based on the analyzed variables).

We rewrote the paragraph according reviewer. The concentration of dissolved oxygen recorded negative changes with Secchi disk visibility (r= -0.896, p≤0.05).

  1. Lines 265-269: Of course, chlorophyll-a does not describe the physico-chemical properties.

We agree with Reviewer so we changed the title of figure 6A and 7A from physicochemical to environmental parameters.

  1. Lines 161-264: According to the horizontally oriented dendrogram in Fig. 6B, the basic division distinguished two groups, the "lower" group comprising rather samples from autumn (Oct - all studied habitats, i.e. BR, NCH and BCH, Sept - BR and NCN) and some summer habitats BR (June, Aug) and NCH (July ). The second group is the spring rehearsals - 6 rehearsals from May and April + Aug from BCH, and the summer rehearsals from both channels. In my opinion, this division indicates the greatest differentiation of zooplankton in the Bydgoszcz Canal - the largest number of samples (Sept and Aug) was outside the groups consisting mainly of samples from spring, summer and autumn.

We changed description of dendrogram 6B according reviewer suggestion.

  1. Lines 280-281: I consider both triplots to be the most important at work, but the question arises whether all the graphs previously posted are actually necessary? Fig. 7a shows that in terms of physicochemical properties and chlorophyll content, the water in Brda is significantly different from the water in all canals, but also that the properties in the Okole site are significantly different from those found at other sites in the Bydgoszcz canal . (You can already see it in Table 1.) However, a question arises - was it correct to carry out the calculations on the average values calculated for all positions in the Bydgoszcz Canal? In my opinion, the answer is unequivocal - NO (especially at the stage of CCA analyzes). It is surprising, however, that it is not the zooplankton at the Okole site on the 7B triplot that is more similar to the one from the Brda and the Notec Channel. It turns out that the zooplankton in the Osowa site appears to be the most similar to the zooplankton in the Brda River and the Notec Channel.

The CCA figures are statistically important, because they show us dependencies between environmental and biological parameters in each studied site separately. We compared in CCA statistics all environmental variables with all biological variables in Fig 3 on Bydgoszcz Canal, Fig 4 on Notec Canal and Fig 5 on Brda River. We used the CCA statistics because it allows to compare more than two variables. So we configured all possible environmental variables with biological data. There is a zooplankton at the Okole site in figure 7B (black circles meant that there was the most zooplankton – matrix coding max.) at Notec Canal and Brda River was less zooplankton (white cycles meant at least zooplankton – matrix coding min.). At Osowa Gora site was average number of zooplankton (more than in Notec Canal and Brda River but less than in Bydgoszcz Canal sites).

  1. Lines 304-308 and 313-319: Much of this information is not found in the Materials and Methods section, and therefore where it should be.

Line 305-308, 310-314, 318-322 was removed to Materials and Methods section. However, line 314-319 with citations are suitably placed in discussion.

  1. Lines 376-378: The ordering diagram does not show that rotifers abundance and biomass correlate with chlorophyll and DO only in spring - it shows that they changed similarly to temperature and DO.

According newly prepared triplot for Bydgoszcz Canal the results showed that biomass of rotifers and their abundance had the greatest dependence of changes with the chlorophyll concentration (r=0.792, p ≤0.01), (r=0.748, p≤0.01).

  1. Lines 404-405: Evidently untrue - the zooplankton at the Osowa site was more similar to that of the Brda River and the Notecki Canal than other sites in the Bydgoszcz Canal!

We corrected it.

Reviewer 3 Report

The authors of the present manuscript present the study of the zooplankton structure in three different water systems and its relationship with the physicochemical parameters. It seems like they want to understand the presence of some alien species in these systems and compare it between them. However, this scope seems lost along the course of the manuscript. They hypothesize that the possible reason of their presence might be in the different hydrological regimes between such water bodies. However, the hydrological regime is not measured in the present study. The last paragraph of the introduction should present better the aim of the manuscript. For me, a paragraph like the one they write at the beginning fo the discussion would be more suitable than the one there is at the moment.

However, the main concern for me is that I would expect different zooplankton structures even at different locations along the river. So, the comparison between the two canals and the river seems for me not completely sound. A detailed scope is needed to clearly state the novelty and meaning of the manuscript.

The discussion is a set of disconnected sentences. I think that it needs more cohesivity. It happens the same with the introduction. For example, line 302 and line 303, are an example of disconnection. References to other studies like this in the discussion need to be related to the present manuscript and the results here presented.

Lines 347-354. They present here a hypothesis based on the hydrological regime. But no measurement of the hydrology is presented in this manuscript. Also, the conclusions should be focusing on the findings of the present manuscript and their relevance, and they go again to the hydrological regime hypothesis.

Minor:

Figures 6 and 7. I think they would be clearly seen in color.

Author Response

The authors would like to thank the Scientific Editor for the constructive comments and for the insightful review of second and third Reviewers. We tried to improve the  first CCA figure, results and discussion chapters and we reformulate conclusions. The results and discussion chapters have been partially rewritten. We add information about water flow to bold difference between canals and river. According to Editor suggestions we have improved all tables, and citations of tables in text of MS. We use the professional English proofing service.

Responses to the reviewer's comments - Reviewer #3

  1. The authors of the present manuscript present the study of the zooplankton structure in three different water systems and its relationship with the physicochemical parameters. It seems like they want to understand the presence of some alien species in these systems and compare it between them. However, this scope seems lost along the course of the manuscript. They hypothesize that the possible reason of their presence might be in the different hydrological regimes between such water bodies. However, the hydrological regime is not measured in the present study. The last paragraph of the introduction should present better the aim of the manuscript. For me, a paragraph like the one they write at the beginning fo the discussion would be more suitable than the one there is at the moment.

We measured water flow in the present study. The aim was partially changed. The discussion was rebuilt.

  1. However, the main concern for me is that I would expect different zooplankton structures even at different locations along the river. So, the comparison between the two canals and the river seems for me not completely sound. A detailed scope is needed to clearly state the novelty and meaning of the manuscript.

We wanted to compare environmental condition including hydrological character between two canals and natural river, thus we took one sampling point from river as an example material. We wanted to demonstrate different water flows in those water basins e.g. Bydgoszcz Canal with the slowest water flow, Notec Canal – untypical canal with faster water flow (The canal that flows in the middle of the Bydgoszcz Canal and may have an influence on the conditions in it). and Brda River with the fastest water flow.

  1. The discussion is a set of disconnected sentences. I think that it needs more cohesivity. It happens the same with the introduction. For example, line 302 and line 303, are an example of disconnection. References to other studies like this in the discussion need to be related to the present manuscript and the results here presented.

The line 302-303 was deleted. Discussion sequence was changed according results sequence. Introduction is clearly presented.

  1. Lines 347-354. They present here a hypothesis based on the hydrological regime. But no measurement of the hydrology is presented in this manuscript. Also, the conclusions should be focusing on the findings of the present manuscript and their relevance, and they go again to the hydrological regime hypothesis.

In our manuscript was measured water flow, conclusions were changed in accordance with the results. We were not able to use the hydrological data in statistical analyzes because we did not have the complete flow results (for all growing season).

  1. Figures 6 and 7. I think they would be clearly seen in color.

Unfortunately, the program we used does not create colorful figures.

Round 2

Reviewer 2 Report

The manuscript was significantly improved. The discussion and conclusions were changed, and the description of research methods and positions was supplemented with the information indicated in the review, previously dispersed throughout the text. The authors took into account all suggested changes. The current version is definitely better. Nevertheless, in the text, despite being subjected to professional linguistic verification, you can come across strange or imprecise phrases, as in the examples given below.

- The expression on lines 217, 244 and 264 "The results of triplot at [...] showed that ...", that is, "The results of the graph show?" - this needs to be corrected by deleting either "results" or "triplot". After all, the results of the CCA are discussed.

- In lines 377-378 it is stated that "rotifer biomass decreased (due to the impact of temperature and macrophytes development)". At this point, the cited literature should be cited, because, despite the changes in the text, the question of the dependence of zooplankton abundance on macrophyte development was not analyzed in the study. The description of the sites only shows that "the Bydgoszcz Canal ... was almost completely covered with floating vegetation and partly with submerged vegetation (lines 88-90). In the description of the remaining sites, nothing was mentioned about macrophytes.

Only in lines 445-447 one can read: "Differences between the Bydgoszcz Canal, Noteć Canal and Brda River occurred throughout the growing season and were probably due to [...] and the different levels of macrophyte vegetation.

- The sentence in lines 398-399: “The results of CCA showed a similar relationship between the Bydgoszcz Canal and the Noteć Canal (Figure 3, Figure 4).” - This sentence does not reflect the proper meaning of the results obtained. They show that in these channels similar relationships between the environmental conditions and the structure of zooplankton were found. Therefore, to say that a relationship has been demonstrated between these habitats is perhaps too far a generalization.

- In the last paragraph "Conlusion" (lines 471-473) it should be emphasized that the lack of "... relationship [...] between the content of nutrients and the production of primary algae (chlorophyll)" concerns this environment (Brda River), thus This statement is not of a general nature. - E.g.: “The analysis of the Brda River did not show any relationship between changes in SD and oxygen concentration in this habitat, and between the nutrient content and the production of primary algae (chlorophyll)”.

Author Response

We would like to thank Reviewer # 2 for the constructive comments and for the insightful review.  According to suggestions we tried to improve the strange or imprecise phrases in text.  

The expression on lines 217, 244 and 264 "The results of triplot at [...] showed that ...", that is, "The results of the graph show?" - this needs to be corrected by deleting either "results" or "triplot". After all, the results of the CCA are discussed.

It was corrected.

- In lines 377-378 it is stated that "rotifer biomass decreased (due to the impact of temperature and macrophytes development)". At this point, the cited literature should be cited, because, despite the changes in the text, the question of the dependence of zooplankton abundance on macrophyte development was not analyzed in the study. The description of the sites only shows that "the Bydgoszcz Canal ... was almost completely covered with floating vegetation and partly with submerged vegetation (lines 88-90). In the description of the remaining sites, nothing was mentioned about macrophytes.

Only in lines 445-447 one can read: "Differences between the Bydgoszcz Canal, Noteć Canal and Brda River occurred throughout the growing season and were probably due to [...] and the different levels of macrophyte vegetation.

We added citation. Indeed, the sites outside the Bydgoszcz canal were poor in macrophytes. 

- The sentence in lines 398-399: “The results of CCA showed a similar relationship between the Bydgoszcz Canal and the Noteć Canal (Figure 3, Figure 4).” - This sentence does not reflect the proper meaning of the results obtained. They show that in these channels similar relationships between the environmental conditions and the structure of zooplankton were found. Therefore, to say that a relationship has been demonstrated between these habitats is perhaps too far a generalization.

We improved it. 

- In the last paragraph "Conlusion" (lines 471-473) it should be emphasized that the lack of "... relationship [...] between the content of nutrients and the production of primary algae (chlorophyll)" concerns this environment (Brda River), thus This statement is not of a general nature. - E.g.: “The analysis of the Brda River did not show any relationship between changes in SD and oxygen concentration in this habitat, and between the nutrient content and the production of primary algae (chlorophyll)”.

It was changed according suggestion.

Reviewer 3 Report

I have read the new version of the manuscript. Authors have done a considerable effort and the manuscript has been improved in the details, the discussion and the presentation of the results. However, I still do not see the scope of the work. They compare the composition of zooplanktonic organisms in two canals and one river. In one canal they have 5 sites whereas in the other systems they have only one site. What is the purpose of the comparison between them when this composition can vary along one of these systems only? So, considering this, has this comparison any meaning?

I also see that authors have not given as much stregnth to the hydrology, that is not measured in the study. They only point the possible differences between the hydrology of the three systems.

Author Response

We would like to thank Reviewer # 3 for the constructive comments and for the insightful review. 

I have read the new version of the manuscript. Authors have done a considerable effort and the manuscript has been improved in the details, the discussion and the presentation of the results. However, I still do not see the scope of the work. They compare the composition of zooplanktonic organisms in two canals and one river. In one canal they have 5 sites whereas in the other systems they have only one site. What is the purpose of the comparison between them when this composition can vary along one of these systems only? So, considering this, has this comparison any meaning?

Thanks a lot for your comments. In the early first version of the text, the title emphasized that these were initial, case study research. Therefore, some shortcomings may have arisen. At the moment, we are continuing the research by carefully measuring the hydrological data, and in addition, on selected locks, we observe the influence of the barrier on the environmental and biological conditions before and after hydrotechnical structures.

I also see that authors have not given as much stregnth to the hydrology, that is not measured in the study. They only point the possible differences between the hydrology of the three systems.

According to reviewer suggestions now we will put much more effort into measuring and interpreting hydrological data influence on zooplankton in various canals.